# Novel Incremental Conductance Feedback Method with Integral Compensator for Maximum Power Point Tracking: A Comparison Using Hardware in the Loop

**Sérgio André** [1,2,3,*] , **Fernando Silva** [1,3,*] , **Sónia Pinto** [1,3] and **Pedro Miguens** [2,3]

1 Departamento de Engenharia Eletrotécnica e de Computadores, Instituto Superior Técnico, University of Lisbon, Av. Rovisco Pais 1, 1049-001 Lisbon, Portugal

2 Instituto Superior de Engenharia de Lisboa, Instituto Politécnico de Lisboa, R. Conselheiro Emídio Navarro, 1959-007 Lisbon, Portugal

3 Instituto de Engenharia de Sistemas e Computadores, Investigação e Desenvolvimento em Lisboa (INESC-ID), R. Alves Redol 9, 1000-029 Lisbon, Portugal

* Correspondence: sergio.a.andre@tecnico.ulisboa.pt (S.A.); fernando.alves@tecnico.ulisboa.pt (F.S.)

**Abstract:** Research on renewable energy sources and power electronic converters has been increasing due to environmental concerns. Many countries have established targets to decrease $CO_2$ emissions and boost the proportion of renewable energy, with solar power being a prominent area of investigation in the recent literature. Techniques are being developed to optimize the energy recovered from PV cells and increase system efficiency, including modeling PV cells, the use of converter topologies to connect PV systems to high-power inverters, and the use of MPPT methods. Certain MPPT algorithms are intricate and demand high processing power. The literature describes several MPPT methods; however, the number of hardware resources required by MPPT algorithms is typically not disclosed. This work proposes a novel MPPT technique based on integral feedback conductance and incremental conductance error, considering the current dynamics of the boost converter. This MPPT algorithm is compared to the most widely used techniques in the literature and evaluates each method's efficiency, performance, and computational needs using an HIL system. Comparisons are made with well-known MPPT algorithms, such as perturb and observe, incremental conductance, and newer techniques based on fuzzy logic and neural networks (NNs). As the NN that is most widely used in the literature depends on irradiation and temperature, an additional NN that is trained using the proposed method is also investigated.

**Keywords:** MPPT; HIL; photovoltaic; integral feedback conductance

## 1. Introduction

The adoption of renewable energy production technologies has grown over the past few decades due to more demanding environmental metrics, resulting in the need to reduce $CO_2$ emissions from fossil fuels. As a result, researchers are continuously seeking new methods of maximizing the power production of solar panels [1,2]. Using wide-bandgap semiconductors, such as silicon carbide or gallium nitride [3], and lowering computing demands [4] are two further ways to improve the effectiveness of renewable energy systems.

Tracking the maximum power point (MPP) in a PV panel can be difficult due to the non-linearity of voltage vs. current in PV cells, and the variable response in their output power as a function of irradiance, temperature, solar incidence angle, and output load [5].

MPPs can be located using a variety of techniques [6,7], and can be grouped into offline and online maximum power point tracking (MPPT) algorithms [8]. Offline techniques, which often require briefly disconnecting the PV from the load, are simple but less effective, and are not considered in this work. Online MPPT techniques provide continuous MPP monitoring under a variety of conditions, including different temperatures, solar incidence

angles, and irradiance levels, and do not require the PV panels to be disconnected. Due to its simplicity and low computational requirements, perturb and observe (P&O) is widely used in industry and highly researched in academia [9–13]. Another well-known MPPT method is incremental conductance (INC) [14]; its search algorithm is still being researched in the literature, with authors proposing some modifications [15,16].

Growing low-cost computing power led to the development of more complex controllers in the 1990s and 2000s, such as fuzzy logic (FL) [5,17], neuro-fuzzy controllers [18], artificial neural networks (ANNs) [19–22], and reinforcement learning [23]. Recently, several algorithms have been proposed for addressing the partial shading problem in solar energy systems. These algorithms include those based on flower pollination [24,25], particle swarm optimization [26], and fireworks algorithm-based [27] methods that provide better tracking in cases of sudden changes in irradiation. In fact, the algorithm proposed in this work could be integrated with some of these methods or use array reconfiguration [25,28] to improve performance under partial shading conditions, although partial shading is not specifically investigated in this study.

Still, there are multiple ways in which MPPT algorithms are implemented in low-computation devices [29–33]. However, a comparative study of the hardware resources needed for implementing them could not be found in the literature.

In this study, a new MPPT method is proposed that combines the incremental conductance concept with an integral compensator (IC-INC). This novel approach overcomes the constant increment or decrement value associated with traditional MPPT algorithms, such as P&O and INC, which can limit their effectiveness. By using linear feedback theory to obtain the MPP, the proposed IC-INC MPPT method eliminates this problem and offers simple implementation in a digital context that does not require many hardware resources.

The use of an integral controller to regulate incremental conductance represents a significant contribution to the literature on MPPT. To evaluate the effectiveness of the proposed method, a comparison is made with well-known MPPT algorithms found in the literature, including P&O-, INC-, FL-, and ANN-based approaches using irradiance and temperature. The proposed IC-INC method is also compared with an ANN trained on a dataset generated from IC-INC. The evaluation is carried out under time-varying irradiance conditions, and the compared methods are analyzed in terms of power extracted from the PV panel, response time, oscillations around the MPP, and hardware resources requirements.

This paper is organized as follows: Section 2 presents the mathematical model and discusses well-known MPPT methods. Section 3 presents the power converter topology for the evaluation of the MPPT methods under study, and the novel incremental conductance with integral compensator (IC-INC) MPPT method. Then, the results obtained through simulation and hardware in the loop (HIL) are presented in Section 4. A discussion of the results is presented in Section 5, and the conclusions and recommendations for future work are presented in Section 6.

## 2. Review of PV Model and MPPT Control Methods

### 2.1. Mathematical Model of Solar Panels

There have been many advancements in PV cell modeling through the years, which have helped to closely reflect the behavior of actual PV cell panels. According to [23], a balance between computational speed and accuracy can be obtained using the current source plus diode model shown in Figure 1 in combination with Equations (1)–(4), and five parameters.

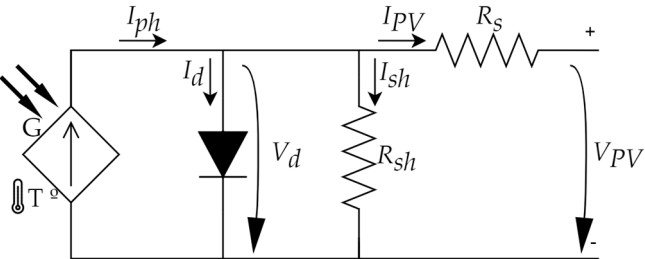

**Figure 1.** Equivalent electrical model of PV cell.

Based on Figure 1, Equation (1) can be written as follows:

$$I_{PV} = I_{ph} - I_d - I_{sh},\tag{1}$$

where $I_{ph}$ represents the current generated by the PV panel under a given solar irradiance $G$ [23].

$$I_{ph} = [I_{sc} + K_l\,(T_c - T_r\,)]\frac{G}{G_{STC}},\tag{2}$$

where

$I_{sc}$—the short circuit current under Standard Test Conditions (STC) (model first parameter)
$K_l$—the short circuit temperature coefficient
$T_c$—the cell temperature
$T_r$—the reference temperature (298.15 K)
$G$—solar irradiance
$G_{STC}$—solar irradiance STC

The diode current in (1) is represented by $I_d$, and in (3), it exhibits exponential behavior [23].

$$I_d = I_0\left[exp\left(\frac{q\,V_d}{A\,k\,T_c}\right) - 1\right],\tag{3}$$

where

$I_0$—the diode saturation current (second parameter of the model)
$q$—the electron charge constant ($1.6 \times 10^{-19}$)
$V_d$—the diode voltage
$A$—the ideal factor of the diode (third parameter of the model)
$k$—Boltzmann's constant ($1.38 \times 10^{-23}$)

The current $I_{sh}$ in (1) is the current in the parallel resistor $R_{sh}$, and can be calculated by:

$$I_{sh} = \frac{(V_{PV} + I_{PV}\,R_s)}{R_{sh}},\tag{4}$$

where

$V_{PV}$—the photovoltaic cell voltage
$I_{PV}$—the photovoltaic cell current
$R_S$—the series resistance of the solar panel (fourth model parameter)
$R_{sh}$—the parallel or shunt resistance of the solar panel (fifth model parameter).

A PV characteristic curve can be obtained using Equations (1)–(4), although choosing the proper model coefficients ($I_{sc}$, $I_0$, $A$, $R_S$, and $R_{sh}$) can be challenging [34]. Figure 2 depicts the characteristic curves of the current and power versus the voltage for the PV used in this study.

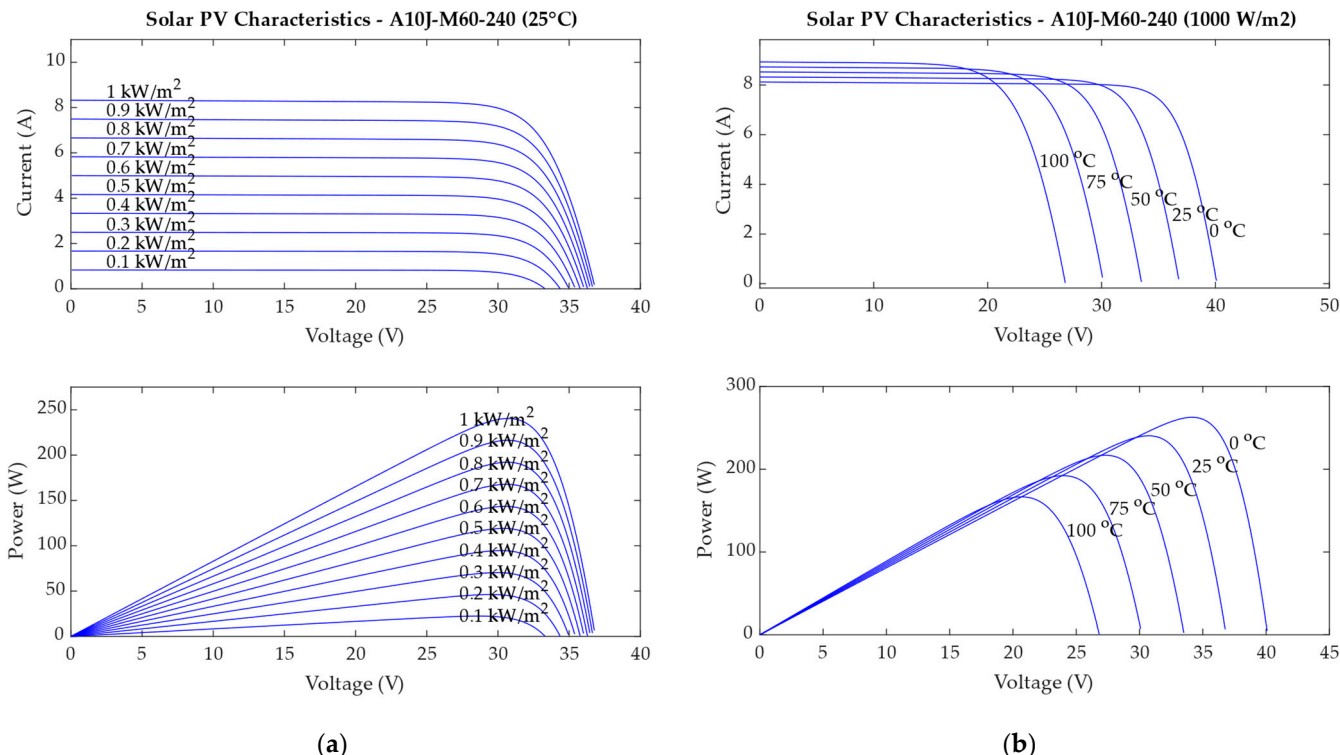

**Figure 2.** Current–voltage and power–voltage characteristic curves of PV A10J-M60-240: (**a**) different irradiances at 25 °C; and (**b**) different temperatures at 1000 W/m².

The PV coefficients of A10-M60-240 from A10Green are already implemented in MATLAB, derived from a PV list based on data from the National Renewable Energy Laboratory (NREL) [35]; however, the PV coefficients are not disclosed by the manufacturer. Instead, manufacturers sometimes provide other characteristics, such as $V_{oc}$ (open circuit voltage), $I_{sc}$ (short circuit current), $V_{MPP}$, $I_{MPP}$, and voltage, as well as the temperature coefficients ($K_v$, $K_i$) [36]. According to [34,37], some methods can be used to model a PV cell when the coefficients are not disclosed.

### 2.2. Review of MPPT Control Methods

MPPT algorithms search for the point of maximum power, represented by the pair ($V_{PV_{MPP}}$, $I_{PV_{MPP}}$), within the plane $V_{PV}$, $I_{PV}$ containing the solar panel's voltage and current curves [7]. Figure 2 illustrates how these curves are non-linear and significantly influenced by cell temperature and sun irradiation [1].

MPPT algorithms are generally tested under specific and often simplistic irradiance conditions [10]. In most scientific papers, only step variation in irradiance is applied; nonetheless, common MPPT algorithms lose their bearings under gradual irradiance variation. Therefore, to more accurately compare the approaches of interest, various patterns of irradiance variation are used in this article.

The following MPPT algorithms are taken into account in the comparison.

#### 2.2.1. Perturb and Observe (P&O)

Due to its low computational complexity, the P&O technique is a straightforward MPPT algorithm and is likely the most used [9–13].

The implementation of P&O is performed in single-loop or multi-loop [38]. In single-loop, the algorithm acts directly on the duty cycle of the electronic power converter, while in multi-loop, the algorithm controls the reference voltage or current. The outcomes in both scenarios are comparable. This paper focuses on the outcomes obtained using single-loop P&O.

The flowchart in Figure 3 depicts the basic form of the P&O method. The algorithm computes the differences between the power and voltage of two samples at a given time, and decides to increase or decrease the duty cycle $D$ of the downstream converter by a constant amount $\Delta D$ in order to iteratively converge into the MPP [11].

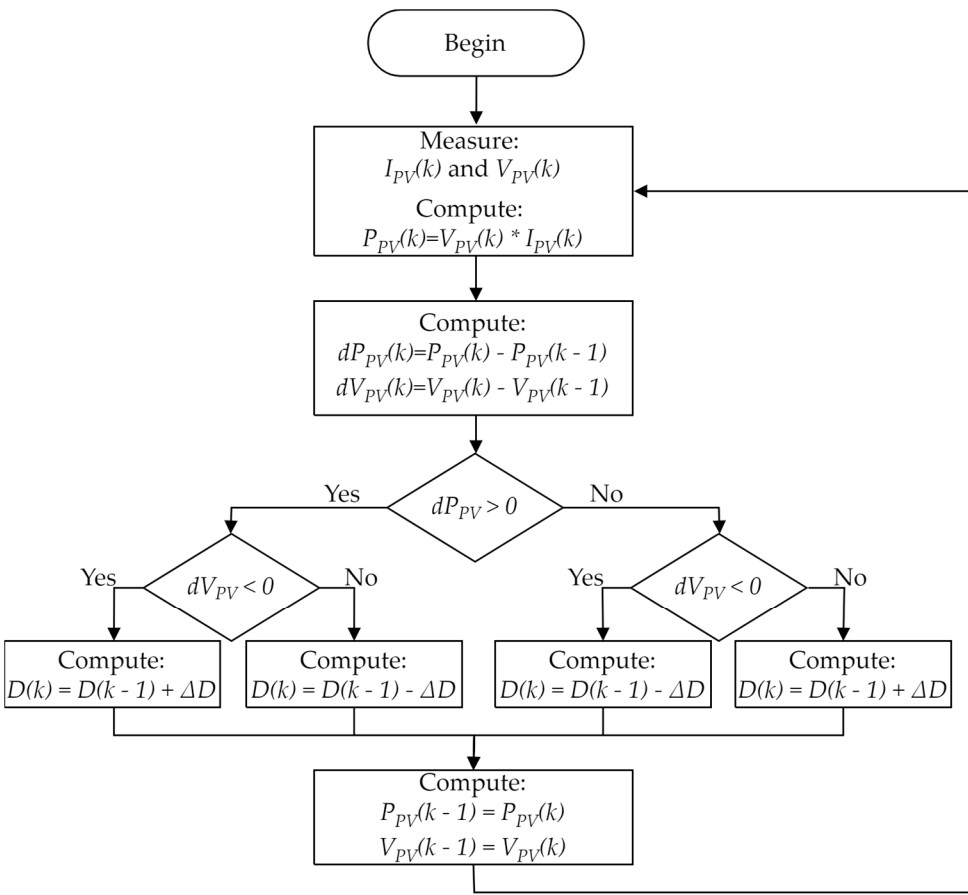

**Figure 3.** Flowchart of P&O algorithm.

In practice, the system never reaches the operating MPP; instead, it oscillates around it with a resolution related to the duty-cycle variation $\Delta D$. It should be noted that a larger $\Delta D$ causes higher perturbation and, as a result, higher oscillation of the voltage and current, whereas a lower $\Delta D$ causes a lower oscillation but slows the response speed [12].

Aside from the undesirable oscillations around the MPP, another disadvantage of this algorithm occurs when the irradiance gradually increases [10]. In these cases, the P&O algorithm temporarily loses the MPP, resulting in power loss.

Some improvements to this algorithm have been proposed, such as the use of adaptive steps or modified flowcharts [9,13]. In this work, a version of P&O with fixed steps is considered.

### 2.2.2. Incremental Conductance (INC)

Another well-known method that is prevalent in the literature is incremental conductance, which is based on the control objective given in (5):

$$\frac{dP_{PV}}{dV_{PV}} = 0, \tag{5}$$

It is clear from the usual *P–V* curve that the system operating point is on the left side of the MPP when the $dP_{PV}/dV_{PV}$ is greater than zero. The operational point is on the right side of the MPP when the value $dP_{PV}/dV_{PV}$ is negative.

From (5), considering the power (given as $P_{PV} = V_{PV}I_{PV}$), (6) is obtained. This result relates the PV panel conductance to its incremental value.

$$\frac{dI_{PV}}{dV_{PV}} = -\frac{I_{PV}}{V_{PV}}, \tag{6}$$

As a result, the duty cycle should be increased if $dI_{PV}/dV_{PV} > -I_{PV}/V_{PV}$. However, the duty cycle should be reduced when $dI_{PV}/dV_{PV} < -I_{PV}/V_{PV}$. At the optimal point $dI_{PV}/dV_{PV} = -I_{PV}/V_{PV}$, the duty cycle is maintained to reduce the ripple in panel voltage and current. This algorithm can be seen in Figure 4.

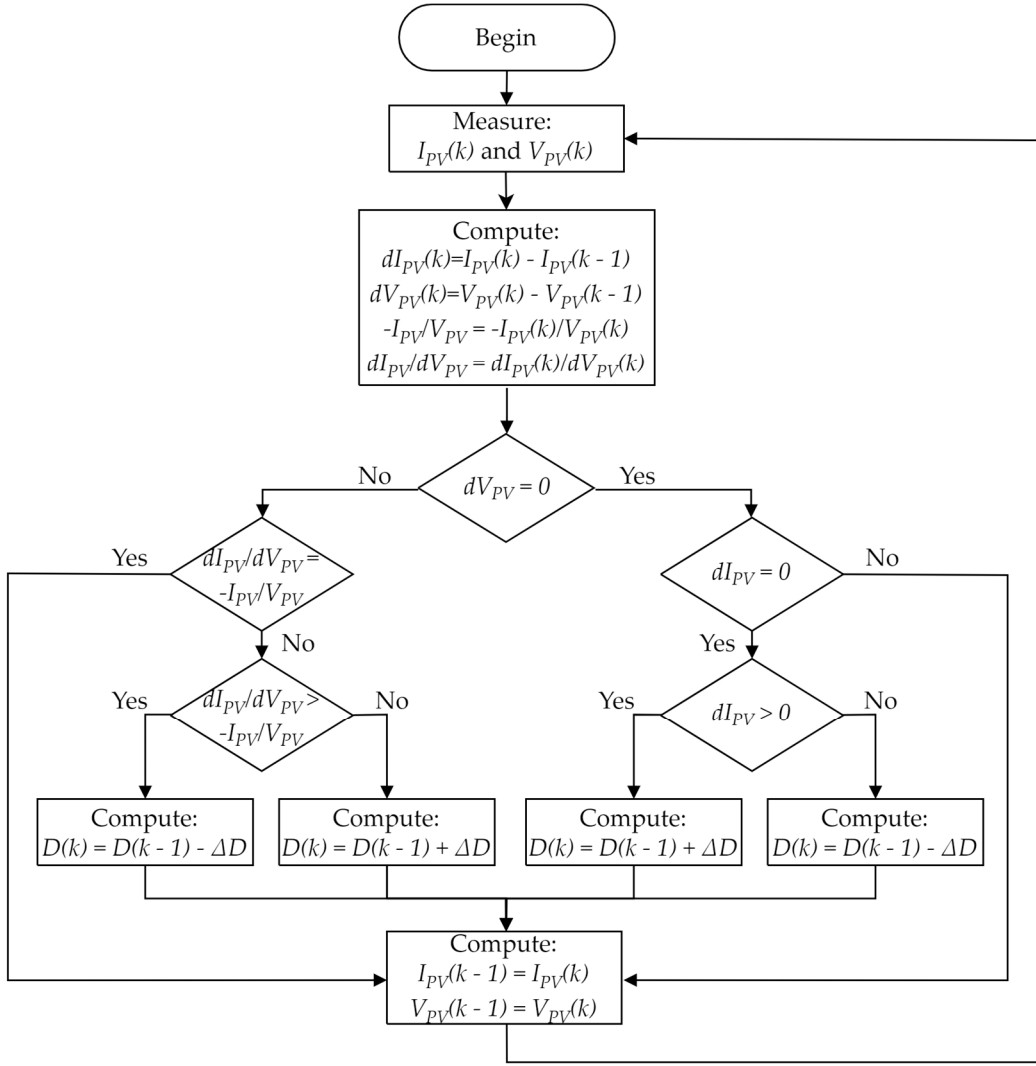

**Figure 4.** Flowchart of incremental conductance algorithm.

Incremental conductance also faces issues when the irradiance changes. During some irradiance changes, the algorithm may compute the $dI_{PV}/dV_{PV}$ value with an incorrect signal, resulting in a delayed transient reaction and, as a result, a momentary loss of power [39].

To lessen issues related to variations in irradiance and transient responses, some adaptive and modified variants of incremental conductance have been introduced [15,16].

2.2.3. Fuzzy Logic (FL)

The fuzzy logic (FL) mathematical framework is often used in artificial intelligence and control systems. Fuzzy logic uses linguistic variables and human expert rules, designed from the expert knowledge of system dynamic behavior, to control complex systems [40]. Therefore, FL does not usually need a mathematical model of the system to be controlled. Instead, FL needs a system dynamics behavior expert to devise a set of required membership functions, linguistic variables, and linguistic rules that translate the expert knowledge of the system dynamics.

In the literature, there are several implementations of the FL-based MPPT algorithm [5,17]. In [5], there are six possible solutions for the fuzzy logic implementation of MPPT.

The discrete form of $dP_{PV}/dV_{PV}$, denoted as $S(k)$, and its discrete time derivative $\Delta S(k)$, are used to develop an FL MPPT algorithm [5]:

$$S(k) = \frac{dP_{PV}}{dV_{PV}} = \frac{P_{PV}(k) - P_{PV}(k-1)}{V_{PV}(k) - V_{PV}(k-1)}, \tag{7}$$

$$\Delta S(k) = S(k) - S(k-1), \tag{8}$$

In this research, the implementation of the FL-based MPPT algorithm in [5] was investigated, in which the duty-cycle variation $\Delta D$ output is based on $S(k)$ and on its discrete derivative $\Delta S(k)$, according to the fuzzy surface control presented in Figure 5.

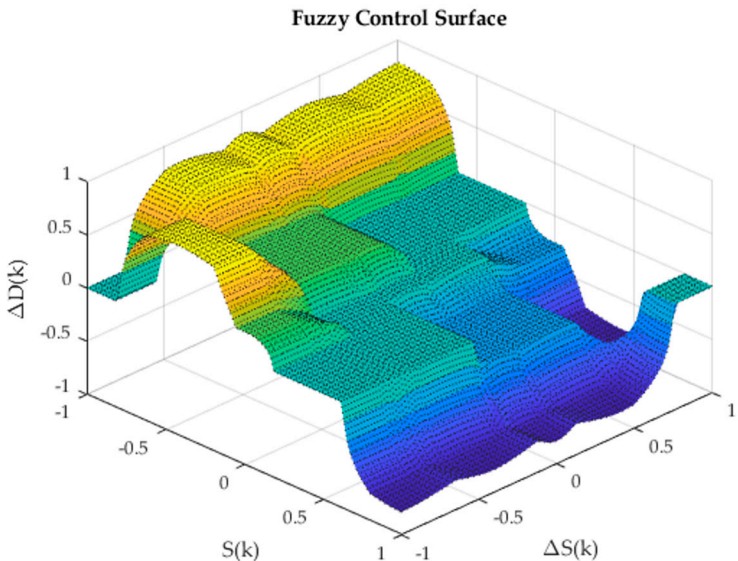

**Figure 5.** Fuzzy logic control surface.

The normalized set of input and output values can be described using five membership functions: negative big (NB), negative small (NS), zero (ZE), positive small (PS), and positive big (PB). Figure 6 shows the input and output membership functions within the normalized universe of discourse. Table 1 shows the devised fuzzy rules. The Mamdani method was employed as an inference engine for FL implementation, and the center of gravity algorithm was used in the defuzzification process.

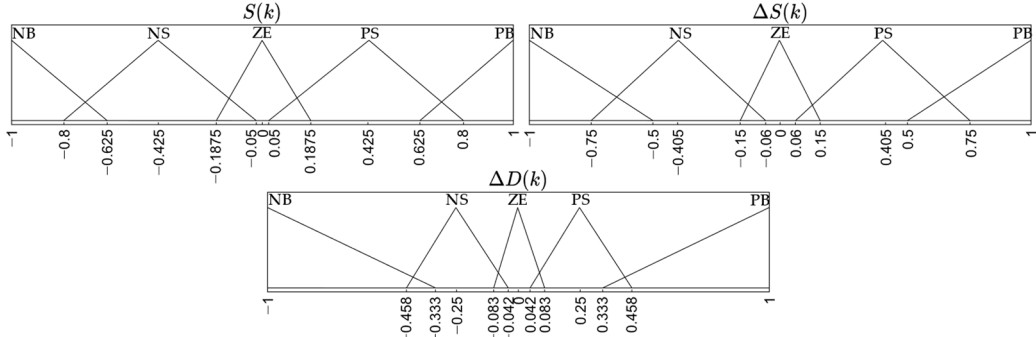

**Figure 6.** Fuzzy logic membership functions.

**Table 1.** Fuzzy rule table based on [5].

| Fuzzy Rule | | S(k) | | | | |
|---|---|---|---|---|---|---|
| | | **NB** | **NS** | **ZE** | **PS** | **PB** |
| | **NB** | ZE | PB | PS | ZE | NB |
| | **NS** | PB | PS | ZE | ZE | NB |
| **ΔS(k)** | **ZE** | PB | PS | ZE | NS | NB |
| | **PS** | PB | ZE | ZE | NS | NB |
| | **PB** | PB | ZE | NS | NB | ZE |

### 2.2.4. Artificial Neural Network (ANN)

An artificial neural network (ANN) is a form of machine learning algorithm that mimics the structure and function of the human brain. Neural networks are built using input layers of cells (neurons), output layers, and several hidden layers of densely interconnected neurons. Each neuron has a variable activation gain, or weight, given by a non-linear function, that is tuned during the learning (training) process. As a result, some neurons are more active than others in the inputs, hidden layers, and outputs, resulting in distinct output signals according to the tuning obtained from the applied training. Therefore, one of the most significant aspects of neural networks is the quantity and quality of the existing training data, which influence the quality of the neural network outcomes after training.

In MPPT, neural networks can be used in a variety of ways [19–22]. They can be trained using irradiance and temperature datasets [19,22] or datasets generated via alternative methods, such as P&O or FL [20,21].

Implementations based on irradiance and temperature are very accurate in simulation, but they have several drawbacks in practice. For each PV model, the neural network must be trained in order to tune the neuron gains. Additionally, irradiance and temperature sensors are expensive and difficult to calibrate [41]. Another disadvantage can occur if the PV is dusty or shadowed while the irradiance sensor is not, or vice versa. In this case, the irradiance- and temperature-trained ANN has reduced performance, since the sensor data do not correspond to the actual panel irradiance.

The work conducted in [21] represents an example of training using the P&O approach, while [20] presents an ANN trained using the fuzzy logic method. These approaches often do not require additional sensors, instead employing the sensors from the DC/DC converter (current and voltage), which contributes to cost savings.

The number of neurons and hidden layers of a neural network is often determined empirically [42], as it depends on the problem's complexity. The number of hidden layers should be balanced, because while more hidden layers can increase the performance of the neural network, too many layers may result in the network reaching a local minimum rather than a global minimum [42].

As approaches based on irradiance and temperature are frequently investigated in the literature [19,22], a neural network trained with a dataset based on these parameters

(ANN (G, T)) was evaluated in this work. The dataset was created using only MPP PV current values, for irradiances between 0 W/m² and 1000 W/m² with steps of 50 W/m², and at temperatures between 0 °C and 80 °C with steps of 5 °C. Although the PV equations produce non-linear I-V and P-V characteristics, the current values in the MPP have a linear relationship with irradiance and temperature (Figure 7), making the problem straightforward. In this case, only one hidden layer with 10 neurons was needed. This implementation achieved good results.

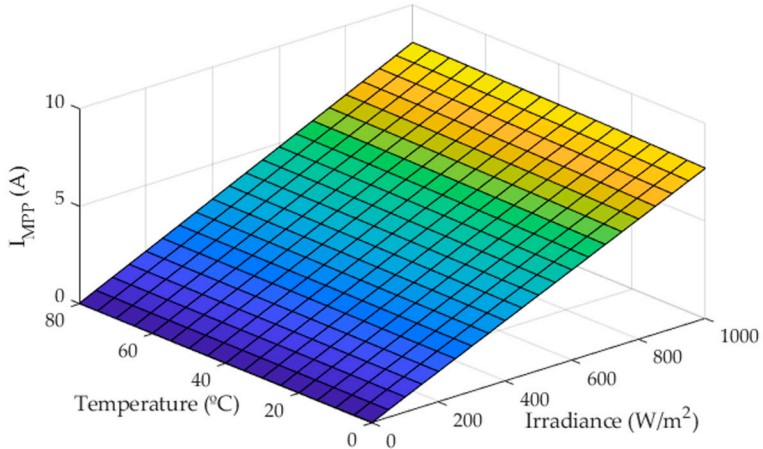

**Figure 7.** Relationship of MPP current with irradiance and temperature.

## 3. Incremental Conductance with Integral Compensator (IC-INC)

### 3.1. Converter Topology

PV panels usually output a relatively low voltage; thus, step-up (boost) converter topologies are frequently used to connect the PV panels to systems that require higher voltage levels, while performing tracking of the MPP [43]. The switching boost converter is connected to a capacitor $C_{in}$ at the PV panel output (Figure 8) to reduce the voltage variation around the MPP. The converter controls the extracted current/power from the PV panel by varying the duty cycle, or by using a hysteretic controller if current control is used. In this work, for ANN and IC-INC, we chose to use hysteretic controllers, while for P&O, INC, and FL, we used a PWM modulator and duty-cycle variation.

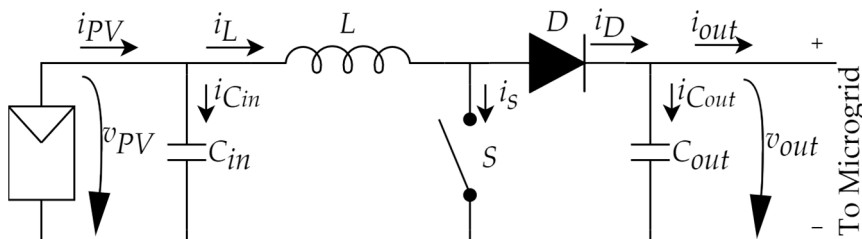

**Figure 8.** Schematic of implemented topology.

The output of the boost converter is connected to a power source ($V_{mg}$) and a resistor $R_{mg}$ that emulates a DC microgrid at 48V, which is typically used in telecom equipment as required by the IEEE DC microgrid standards [44].

The high-frequency $f_{sw}$ switching boost converter stores energy in the inductor $L$ before releasing it in the output capacitor $C_{out}$ at higher voltage. The high-frequency voltage ripple in the PV ($V_{PV}$) should be as minimal as possible, as the MPP is dependent on voltage and current, and oscillations in voltage or current around the MPP can cause a loss of power. Hence, the input capacitor should have enough energy to keep the PV voltage nearly constant during the high-frequency switching period and during abrupt irradiance changes.

An inductor *L* was selected to allow for a variation of 10% in the boost input current. Although the current $i_L$ has a higher variation than $i_{PV}$, both have the same average value ($I_L = I_{PV}$). This variation in currents should be supported by $C_{in}$.

The $C_{in}$ capacitor is appropriately sized for very low ripple voltage (0.05% of $V_{PV}$); thus, the $I_{PV}$ current will also present very little rippling, preventing limit cycles around the MPPT. The $C_{in}$ capacitor keeps the PV voltage and current nearly constant during the switching period.

The output capacitor $C_{out}$ is appropriately sized to support discontinuities in the current while presenting low ripple voltage (0.5%). All the system parameters are presented in Table 2.

**Table 2.** Parameters of implemented topology.

| Parameter | Value |
|---|---|
| $V_{oc}$ (PV open circuit voltage) | 36.84 V |
| $I_{sc}$ (PV short circuit current) | 8.32 A |
| $V_{MPP}$ (PV MPP voltage) | 30.72 V |
| $I_{MPP}$ (PV MPP current) | 7.83 A |
| $P_{MPP}$ (PV MPP power) | 240.54 W |
| $L$ (inductor) | 300 µH |
| $C_{in}$ (input capacitor) | 150 µF |
| $C_{out}$ (output capacitor) | 150 µF |
| $f_{sw}$ (switching frequency) | 50 kHz |
| $V_{mg}$ (microgrid voltage) | 48 V |
| $R_{mg}$ (microgrid resistance) | 50 mΩ |

*3.2. IC-INC Method and Algorithm*

The IC-INC approach is based on the incremental conductance Equation (6), written as (9).

$$V_{PV} \frac{dI_{PV}}{dV_{PV}} + I_{PV} = 0, \tag{9}$$

This equation means that in steady state, the conductance $I_{PV}/V_{PV}$ equals its incremental value $dI_{PV}/dV_{PV}$. However, during transients or during the convergence to steady state, the algebraic sum of the incremental conductance value $dI_{PV}/dV_{PV}$ with the conductance $I_{PV}/V_{PV}$ will not be zero; this will lead to a tracking error $e_{MPPT}$, rewritten in (10) as the error, or deviation, of the negative feedback system.

$$V_{PV} \frac{dI_{PV}}{dV_{PV}} - (-I_{PV}) = e_{MPPT}, \tag{10}$$

This tracking error value $e_{MPPT}$ should be enforced to zero within a finite amount of time using a suitable closed-loop controller. Upon dividing all the components of (10) by $V_{PV}$, (11) is obtained.

$$\frac{dI_{PV}}{dV_{PV}} - \left(-\frac{I_{PV}}{V_{PV}}\right) = \frac{e_{MPPT}}{V_{PV}}, \tag{11}$$

Assuming the boost converter is driven by a current controller (e.g., the hysteresis controller), the inductor current $I_L$ dynamics will track the current reference $I_L \approx I_{L_{ref}} \approx I_{PV}$ with a small delay. This relatively small delay can be approximated using a first-order low-pass filter dynamics with a dominant pole at $-1/Tc$. Then, the term $-I_{PV}/V_{PV}$ in (11) can be written as follows.

$$-\frac{I_{PV}}{V_{PV}} \approx -\frac{I_{L_{ref}}}{V_{PV}} \frac{1}{1 + sTc}, \tag{12}$$

If we consider the capacitor at the output of the PV $C_{in}$ is parallel with an equivalent PV resistor, given by $R_{PV} = V_{MPP}/I_{MPP}$, $T_c$ can be estimated as follows.

$$T_c = C_{in}R_{PV} = C_{in}\frac{V_{MPP}}{I_{MPP}}, \tag{13}$$

To ensure MPPT tracking (zero steady-state error), an integral controller $K_i/s$ can then be devised using (10) and (12) (Figure 9). The integral gain $K_i$ can be computed using the closed-loop transfer function represented in Equation (14).

$$\frac{-\frac{I_{PV}}{V_{PV}}}{\frac{dI_{PV}}{dV_{PV}}} \approx \frac{-\frac{K_i}{T_c}}{s^2 + \frac{s}{T_c} - \frac{K_i}{T_c}}, \tag{14}$$

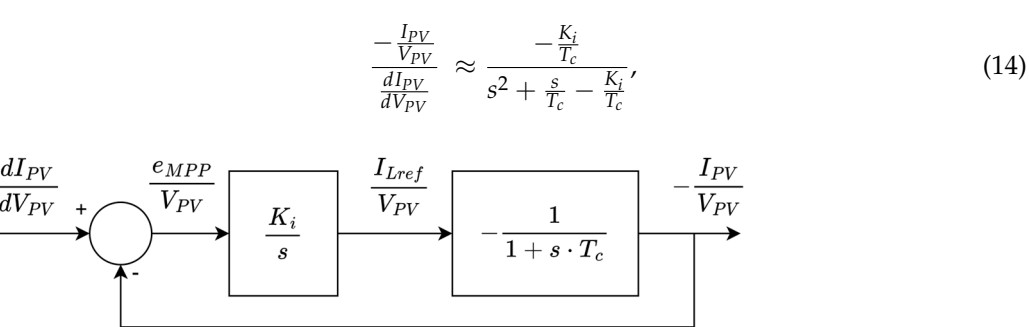

**Figure 9.** Block diagram of the novel integral compensator MPPT method.

Compared with the canonical form of a second-order system $s^2 + 2\xi\omega_n + \omega_n^2$, (15) is obtained. To ensure stability, $K_i$ should be negative with a damping factor close to unity $\xi \approx 1$.

$$\omega_n^2 = -\frac{K_i}{T_c}; K_i = -\frac{1}{2\,\xi^2 T_c}, \tag{15}$$

*3.3. Using the IC-INC Method to Train a Neural Network*

It can be seen that MPPT ANNs trained using irradiance and temperature datasets may present severe drawbacks in practice. To use the voltage and current from PVs, a dataset for training a new neural network is obtained through the IC-INC method applied to the PV.

The new neural network, denoted IC-INC, uses the voltage and current of the PV and their last values ($t-1$) as inputs. The ANN IC-INC outputs the variation in the reference current, an external integrator in the output of the ANN that mimics the integrator in the IC-INC, and additional multiplication by $V_{PV}$ is required (11).

The IC-INC method includes divisions (11) of $I_{PV}$ by $V_{PV}$ and $dI_{PV}$ by $dV_{PV}$. This is a more sophisticated process than learning a surface, as performed by ANNs (G, T), because of the known division operation issues that occur in ANNs. As such, two different ANNs were considered in this paper. The first uses three hidden layers and five neurons in the first and second layers, and three neurons in the third layer, while the second uses four hidden layers and six neurons in the first and second layers, four neurons in the third layer, and two neurons in the fourth layer.

When the results of the two trained ANNs were compared, it was determined that the neural network with four hidden layers provided the best answer, because no over-fitting was discovered during its training phase and its performance was higher. The ANN's performance was measured using two parameters: the mean square error (MSE) and R-squared (R). The first implementation with three hidden layers obtained an MSE of $3.72 \times 10^{-7}$ and an R of 0.85. The implementation with four hidden layers achieved an MSE of $2.48 \times 10^{-7}$ and an R of 0.92.

## 4. Results
*4.1. Laboratory Setup*

The results were obtained through simulation and an HIL test system. The HIL is shown in Figure 10, and consists of FPGA Zynq ZC706 and some interface boards (DACs

and board adapters). The system was inspired by a controlled HIL system (C-HIL), with the controller and the emulated system (PV, boost converter, and DC microgrid) implemented in FPGA ZC706.

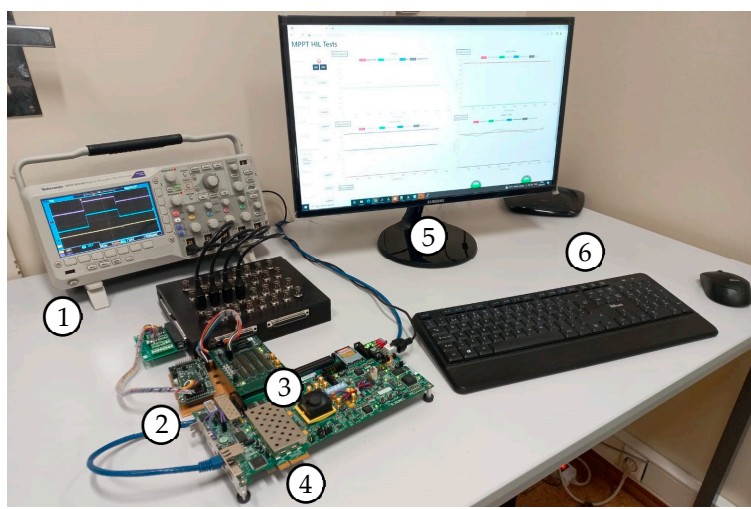

**Figure 10.** MPPT HIL test setup. 1—Oscilloscope; 2—DAC7558EVM; 3—expansion board XM105; 4—FPGA Xilinx ZC706; 5—computer; 6—router.

Emulation of the PV system, boost converter, and microgrid was conducted using Simscape electrical components, followed by conversion into a linear state-space model. This conversion produced four typical state-space matrixes and a Simulink file with implementation of the state space. This file could then be coded into VHDL using an HDL coder in MATLAB. Then, the VHDL code was entered into Vivado from Xilinx (AMD) [45], compiled, and transferred to the FPGA.

As the PV panel in Simscape was not available for the HDL coder, it was replaced by the PV model reviewed in Section 2.1 and presented in Figure 1. This implementation is shown in Figure 11.

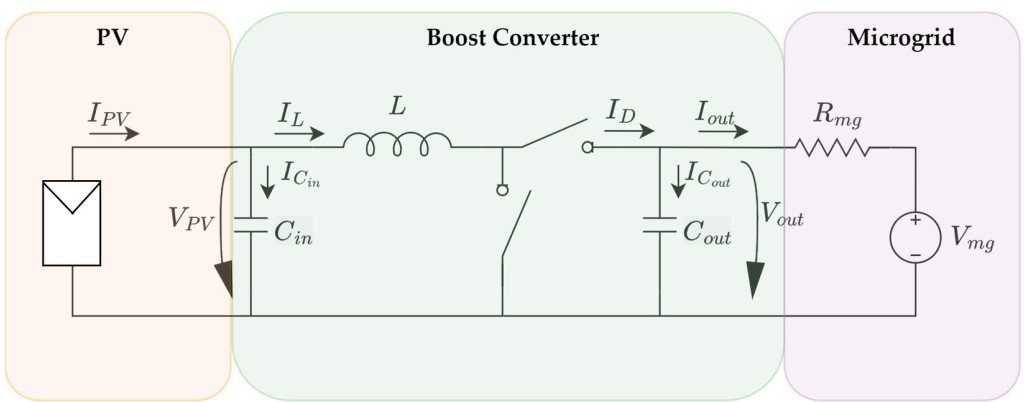

**Figure 11.** HIL implementation diagram.

The VHDL code for the MPPT algorithms was produced using the HDL coder library in MATLAB. This tool requires the usage of single-precision floating point values, and is unable to use double-precision values. To fulfill this requirement, the blocks used in this implementation were configured to work with single-precision values. The use of single instead of double precision reduced resource utilization in the FPGA, but also reduced the precision of the mathematical operations.

After code conversion, the produced VHDL code was added to Vivado in block design mode and connected to a PWM modulator or hysteretic blocks.

For a clearer data visualization, some additional elements were added to Vivado to show the PV power that was instantaneously produced and the maximum available power. These signals were sent to the DAC board to enable visualization using an oscilloscope.

Some additional functions were also added to change the parameters of irradiance and temperature in the HIL model using the FPGA AXI interface. In addition, a webserver was implemented in the Zynq processor to simplify the user's interaction with the system.

The MPPT algorithms were designed in MATLAB Simulink to speed up the implementation process. Moreover, this process enabled simulation of the controller before its implementation.

### 4.2. Test Results

All algorithms were subjected to steady-state tests, step variation in irradiance, and two fast and slow transience scenarios. Variation of 6000 $W/m^2/s$ was considered fast transient variation, while 1000 $W/m^2/s$ was considered slow variation. Each of the algorithms was adapted to obtain maximum power extraction, and thus, an accurate comparison of the results.

The irradiance conditions applied to the PV in the simulation tests are shown in Figure 12, and are described throughout the Results section.

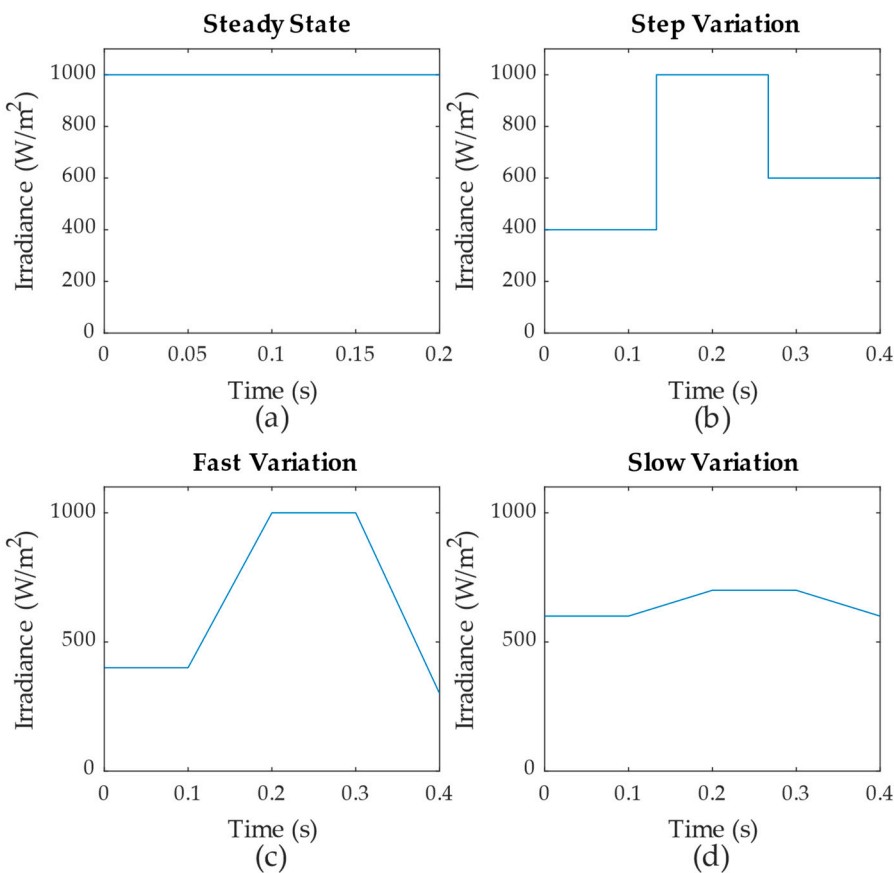

**Figure 12.** Irradiance level applied to PV to simulate different scenarios: (**a**) steady state; (**b**) step variation; (**c**) fast variation; and (**d**) slow variation.

A step variation test was performed to better compare the obtained results with the published ones, as the step test is widely used in the literature. At the beginning of the step test, the irradiance is 400 $W/m^2$; at 0.133 s, this value increases to 1000 $W/m^2$, and at 0.266 s, another transition occurs, with a final value of 600 $W/m^2$. The step variation simulation results of each algorithm are presented in Figures 13 and 14, which contain the HIL results. These results were obtained using a Tektronix DPO 2014B oscilloscope.

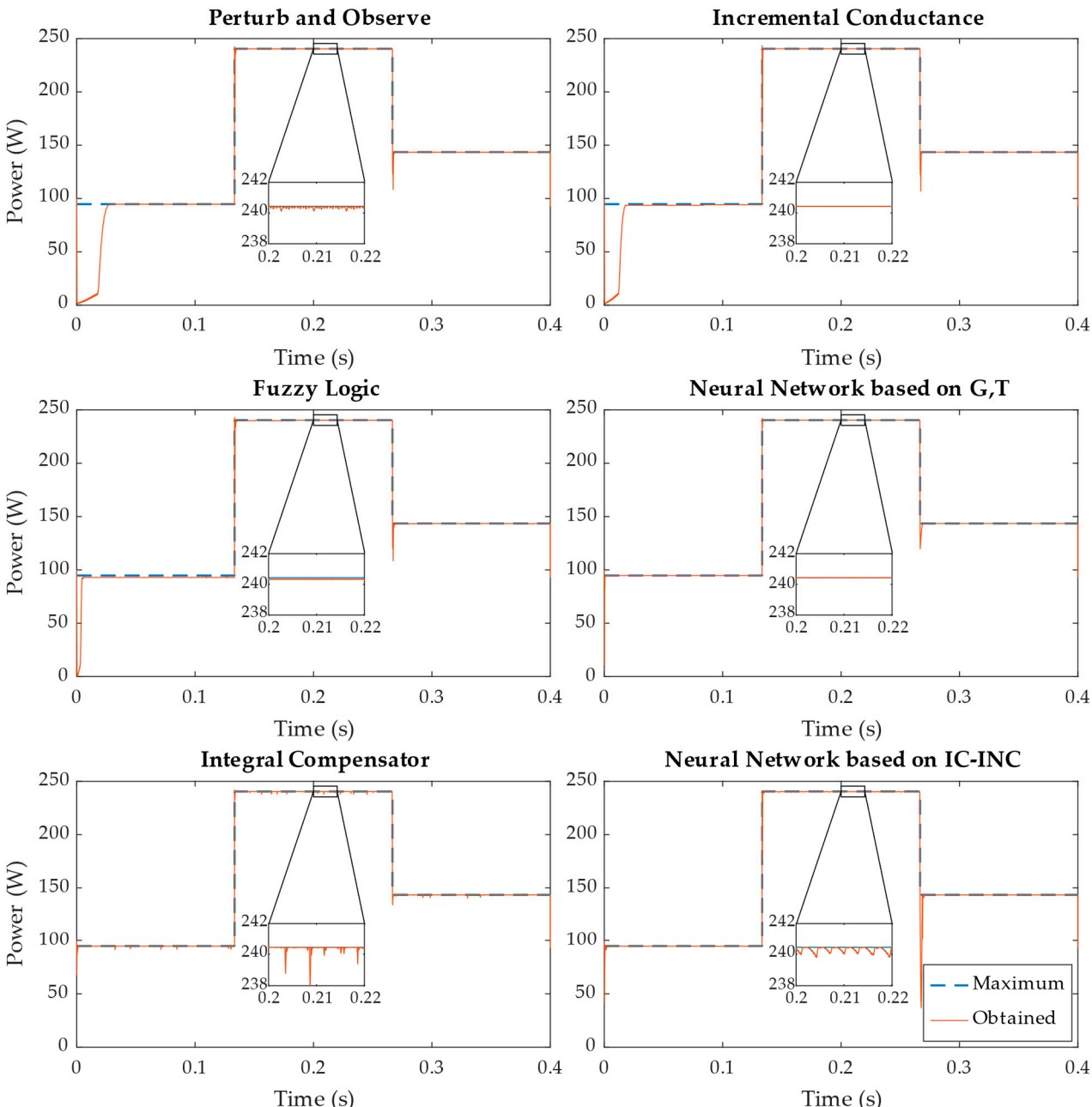

**Figure 13.** Simulation results of power produced from panel when step variations occurred.



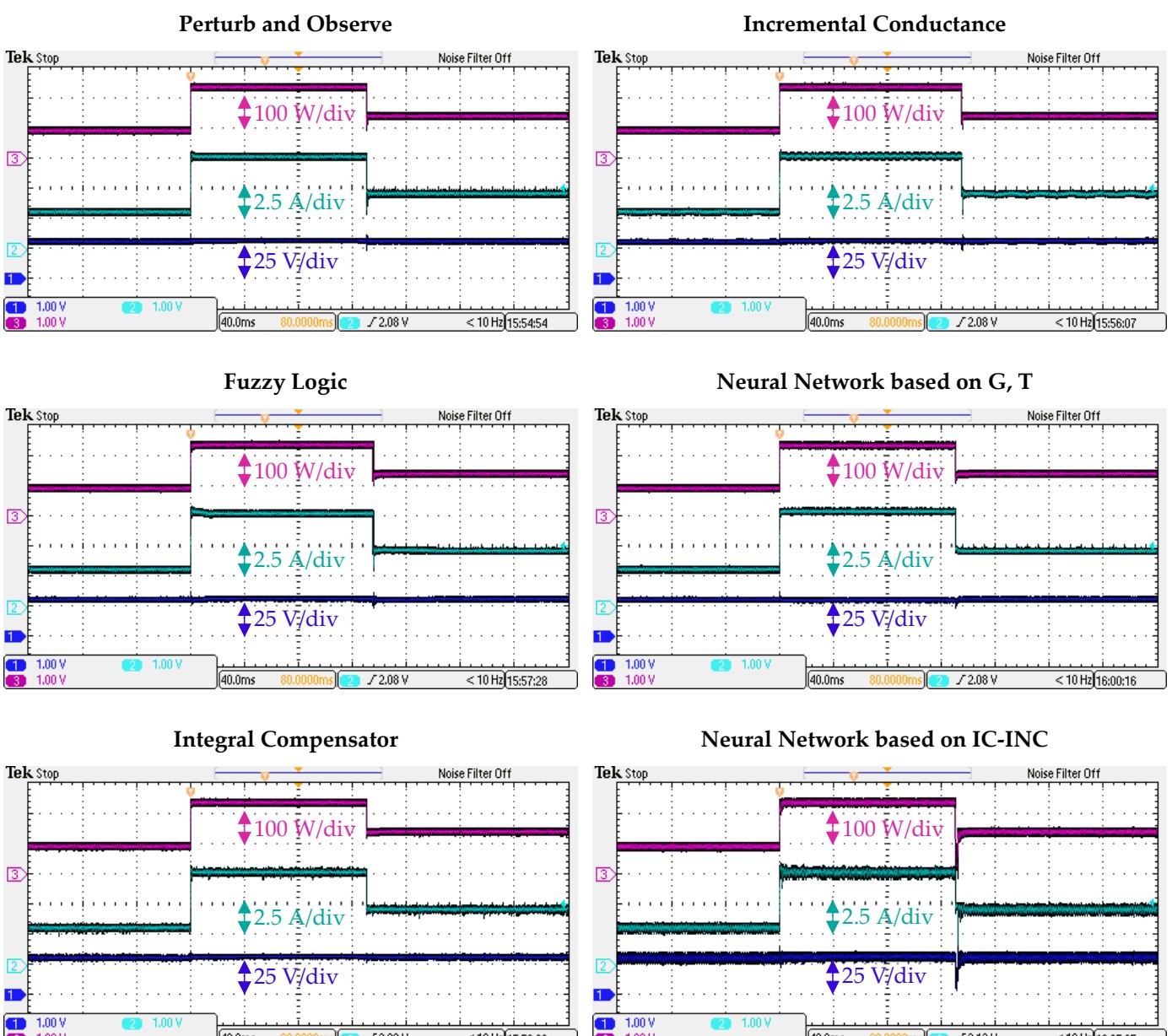

**Figure 14.** HIL results of power produced from panel when step variations occurred.

To better understand the MPPT method's response to linear irradiance changes, fast and slow tests were conducted.

Within the fast test, the initial irradiance value is 400 W/m$^2$; at 0.1 s, the value gradually increases to 1000 W/m$^2$, stabilizes for 0.1 s, and then decreases to 300 W/m$^2$ between 0.3 and 0.4 s. The simulation results for this scenario are presented in Figures 15 and 16, which show the HIL results.

The simulation and HIL results for the slow test are presented in Figures 17 and 18, respectively. The initial irradiance conditions are 600 W/m$^2$ for 0.1 s; then, the irradiance increases from 600 W/m$^2$ to 700 W/m$^2$ for 0.1 s. After this increase, the value remains unchanged for 0.1 s, and finally decreases to 600 W/m$^2$.

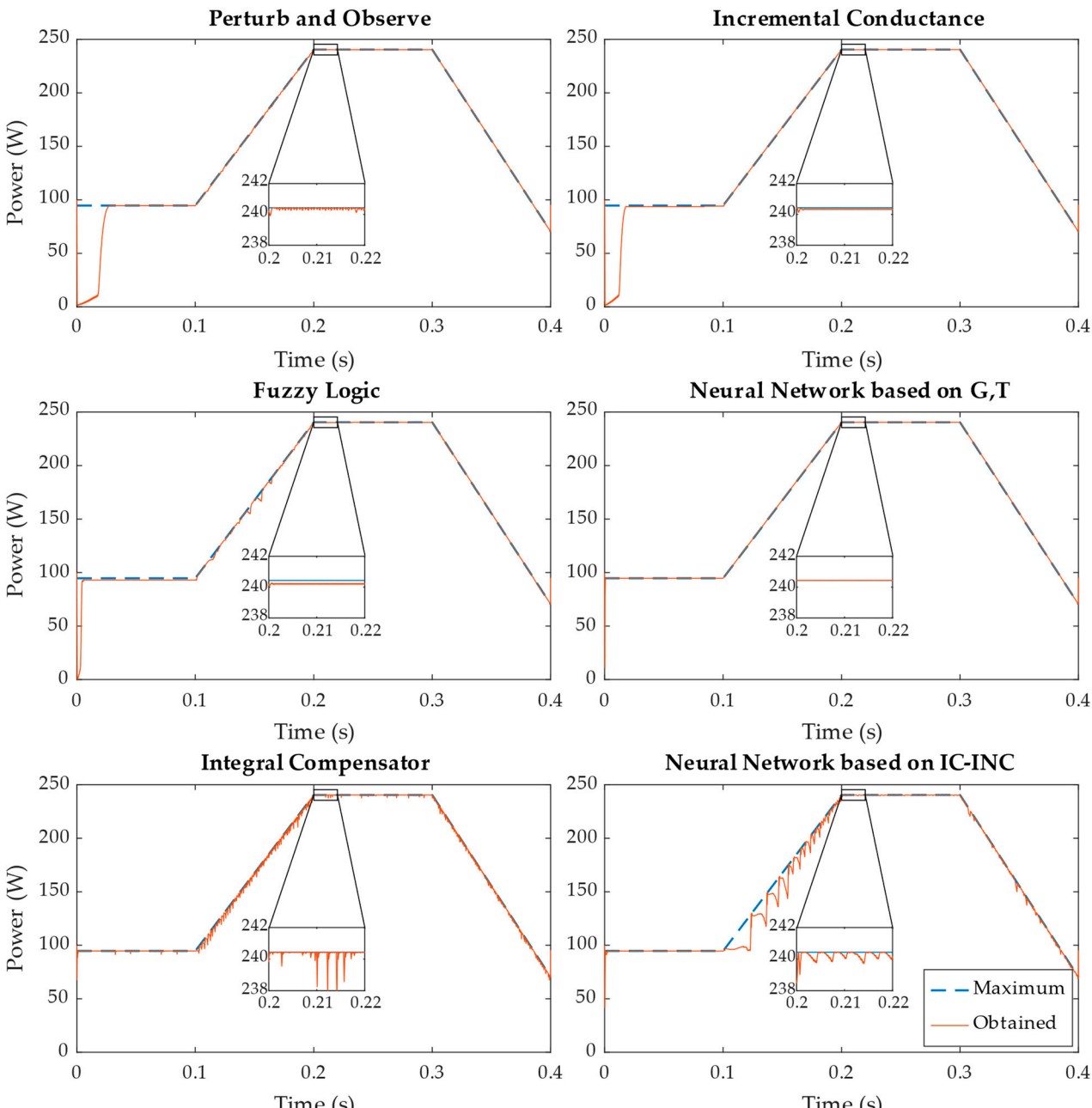

**Figure 15.** Simulation results of power produced from panel when fast variations occurred.

The data in Table 3 complement the presented results, showing the efficiency $P_{MPPT_{Algorithm}}/P_{MPPT}$ in each MPPT algorithm. The efficiencies shown do not include losses in the boost converter (98% efficiency), and only the output power of the PV panel is considered to compute the ratio of the maximum available power. The values presented in Table 3 were obtained using simulation results, and Table 4 presents the results for HIL.

**Table 3.** Simulation results of MPPT efficiency.

| Algorithm | Step Variation | Fast Variation | Slow Variation | Steady State * |
|---|---|---|---|---|
| P&O | 97.03% | 97.14% | 95.58% | 99.98% |
| INC | 97.85% | 98.01% | 96.96% | 99.91% |
| FL | 99.03% | 99.08% | 99.20% | 99.87% |
| ANN (G, T) | 99.94% | **100.00%** | **99.96%** | **100.00%** |
| IC-INC | **99.95%** | 99.67% | 99.89% | 99.97% |
| ANN (IC-INC) | 99.67% | 98.50% | 99.86% | 99.94% |

* Steady-state values were calculated between 0.1 s and 0.2 s, and the initial transient response was discarded.

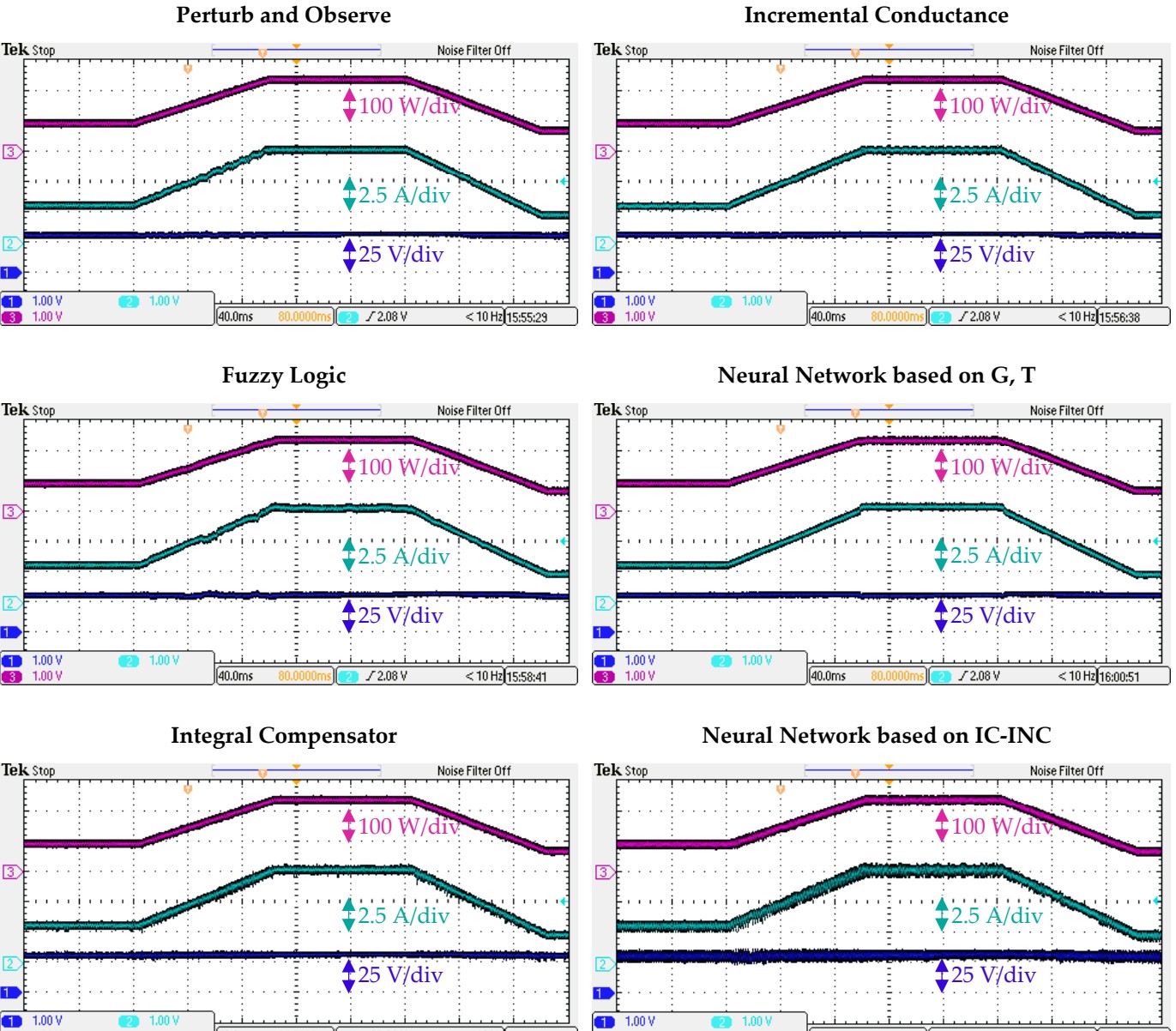

**Figure 16.** HIL results of power produced from panel when fast variations occurred.

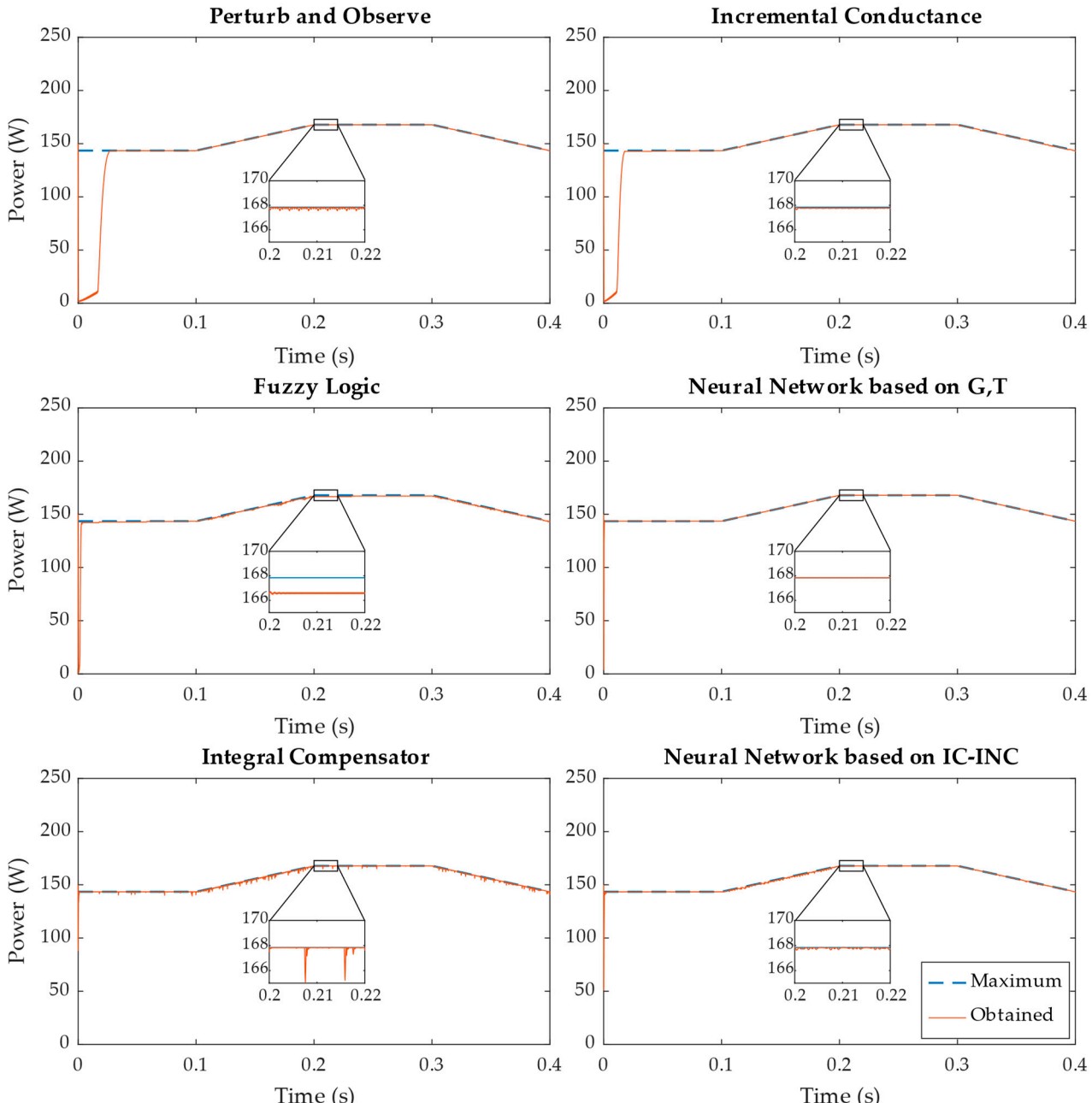

**Figure 17.** Simulation results of power produced from panel when slow variations occurred.

**Table 4.** HIL results of MPPT efficiency.

| Algorithm | Step Variation | Fast Variation | Slow Variation | Steady State |
|---|---|---|---|---|
| P&O | 98.36% | 98.94% | 99.11% | 99.04% |
| INC | **99.21%** | **98.98%** | 99.04% | 99.17% |
| FL | 99.14% | 98.81% | 98.64% | 99.12% |
| ANN (G, T) | 97.88% | 98.55% | 99.06% | 97.95% |
| IC-INC | 98.54% | 98.92% | 99.10% | **99.23%** |
| ANN (IC-INC) | 97.86% | 98.66% | **99.36%** | 99.12% |

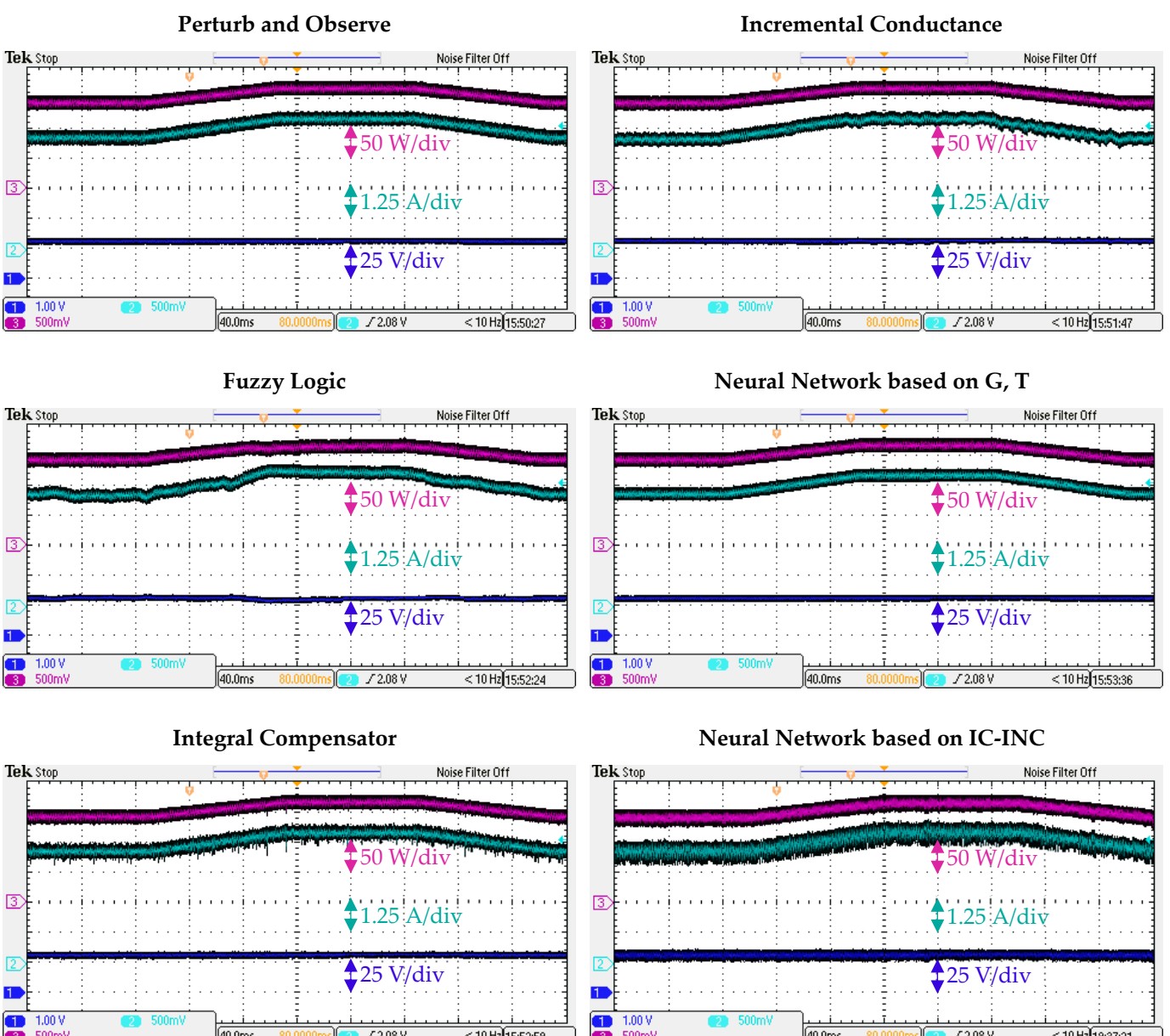

**Figure 18.** HIL results of power produced from panel when slow variations occurred.

*4.3. Required Hardware Resources*

To evaluate the hardware resources required by each MPPT algorithm, a synthesis compilation was performed. The device used was an xc7z045ffg900-2 FPGA from a ZC706 board, and default synthesis configuration was applied. These results are presented in Table 5 as a percentage of occupation in the ZC706 FPGA.

The ZC706 FPGA was divided into two major sections: the processing system (PS) and programable logic (PL). The PS section is related to the ARM processor and is not the focus of the present study. The PL section contained all the implemented algorithms. This section contained 54,650 slices, 900 DSP, and some other dedicated hardware. Each slice contained eight registers, four LUTs, two F7 muxes, and one F8 mux. In total this FPGA contained 218,600 LUTs, 437,200 registers, 109,300 F7 muxes, and 54,650 F8 muxes. The registers are not presented in Table 5 because of their irrelevance in relation to the other components.

**Table 5.** Results of FPGA resources for different implemented algorithms.

| Algorithm | LUTs (218,600) | F7 Muxes (109,300) | F8 Muxes (54,650) | DSPs (900) |
|---|---|---|---|---|
| P&O | 1.08% | 0.01% | 0.00% | 0.22% |
| INC | 2.48% | 0.01% | 0.00% | 0.00% |
| FL | 5.92% | 2.02% | 1.67% | 0.89% |
| ANN (G, T) | 28.38% | 0.08% | 0.02% | 11.78% |
| IC-INC | 2.53% | 0.00% | 0.00% | 0.44% |
| ANN (IC-INC) | 63.76% | 0.08% | 0.02% | 30.67% |

In terms of the hardware resources required for implementation, the cost of the FPGA could be determined for each of the algorithms under consideration. The comparison was conducted using a Xilinx 7 series FPGA, with specific hardware requirements varying between algorithms. Specifically, P&O, INC, and IC-INC require the XC7S6 FPGA from the Spartan family, while FL requires the XC7S25 FPGA. The ANN (G, T) algorithm can be implemented using the XC7S75 FPGA, whereas ANN (IC-INC) requires an Artix XC7A200T FPGA.

Considering only the hardware costs associated with implementing the algorithms, it can be estimated that P&O, INC, and IC-INC would cost approximately EUR 20 when implemented in a microcontroller. On the other hand, FL is estimated to cost twice as much, and ANN (G, T) would require five times more resources for microcontroller implementation. Finally, ANN (IC-INC) is estimated to require 15 times more resources than P&O, INC, and IC-INC in terms of hardware costs.

It is important to note that P&O, INC, and IC-INC can also be implemented in a microcontroller that costs only a few euros. In this case, the hardware resources available would be replaced by CPU usage, with the calculations performed sequentially; however, the cost would be substantially reduced. While FPGA implementation may be necessary for certain applications that require higher performance and greater processing power, such as ANNs, microcontroller implementation represents a more cost-effective solution for many other applications.

## 5. Discussion

The results obtained in this study show that the presented methods respond differently when exposed to different variations in irradiance. Additionally, during steady-state simulations, all algorithms showed effectiveness values exceeding 99%. However, in HIL simulations, the ANN (G, T) exhibits a comparatively lower effectiveness value. Nevertheless, no MPPT algorithm achieves perfect effectiveness. According to our analysis of the response times and oscillations in the voltage and current of the PV, all the algorithms have good response times. In terms of oscillation, the INC, FL, and ANN (G, T) present lower oscillation than the other methods.

According to our analysis of Table 3, the algorithms with the greatest efficiency are the new IC-INC and the ANN (G, T). The IC-INC method is also effective in creating datasets to train high-performance ANNs that are not dependent on measured values of irradiance and PV cell temperature. The simulation results show that FL, IC-INC, and ANN algorithms achieved efficiencies above 99% in all the tested scenarios. In terms of implementation, the proposed IC-INC algorithm does not require a high number of resources, with values slightly higher than P&O and INC.

The fuzzy logic and neural networks algorithms also achieve good results in most scenarios; however, more resources are necessary for FPGA implementation.

The behaviors shown in the results of the HIL and simulation are very similar in most cases; however, in FL implementation, some changes are visible in the slow variation test.

By comparison to the results reported in Table 4, it was observed that the IC-INC algorithm outperforms the tested ones in steady state. Furthermore, the IC-INC algorithm delivered satisfactory results in most irradiance scenarios.

## 6. Conclusions

Based on our results, it is easy to understand the extensive use of the P&O approach in the industry because it is simple, is lightweight, and can obtain results above 99.9% under constant irradiance. As shown in our results, the INC technique, which is also utilized in industry, is an algorithm that uses slightly more resources than P&O, but produces better results than P&O in most tests.

On the other hand, FL- and ANN-based MPPT methods produce even better results (simulation) than P&O and INC; however, they are difficult to deploy because they require expert knowledge or a dataset for training, and many more resources.

The new IC-INC approach appears to be a valuable alternative to P&O and INC, with comparatively better tracking capabilities, a shortened settling time in MPP tracking, and very accurate MPP detection, all while being as light on resources as P&O and INC. Therefore, the proposed algorithm is particularly suitable for use in scenarios where each single PV panel must have its own microconverter to maximize power extraction in difficult conditions, such as partial shading.

Our evaluation of the hardware resources required by each MPPT algorithm shows that P&O, IC-INC, and INC require the fewest computing resources, such as a very low-power microprocessor, while FL- and ANN-based MPPT require many more resources, such as an FPGA or a fast digital signal processor (DSP).

Future research could include testing the promising IC-INC algorithm under partial shading to verify its ability to detect the global MPP, or its potential as a low-resource MPPT algorithm for distributed microconverters in every PV string.

**Author Contributions:** Conceptualization, S.A., F.S., S.P. and P.M.; methodology, S.A., F.S., S.P. and P.M.; investigation, S.A.; writing—original draft preparation, S.A.; writing—review and editing, F.S., S.P. and P.M.; supervision, F.S., S.P. and P.M. All authors have read and agreed to the published version of the manuscript.

**Funding:** This work was supported, in part, by funds from Portugal's Fundação para a Ciência e Tecnologia (FCT) (project reference UIDB/50021/2020) and an FCT PhD grant (reference 2022.12809.BD).

**Data Availability Statement:** Not applicable.

**Conflicts of Interest:** The authors declare no conflict of interest.

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
