# Peer review of "Novel Incremental Conductance Feedback Method with Integral Compensator for Maximum Power Point Tracking: A Comparison Using Hardware in the Loop"

_applsci, doi:10.3390/app13074082_

Round 1
Reviewer 1 Report
The author describes the “Novel Incremental Conductance Feedback Integral Method for Maximum Power Point Tracking: A comparison using Hardware in the Loop”
Ø The paper is well described and can be accepted for the publication
Ø Check the figure number
Ø Ratings of the proposed system with voltage, current, and power should be present for clear understanding.
Ø Conclusions should include the theoretical and experimental efficiency of the system to validate the advantage of the proposed technique.
Author Response
Dear reviewer thank you for all your valuable contributions and suggestions.
Please see the attachment with the response to your comments.
Best Regards,
Sérgio André

Reviewer 2 Report
Authors have proposed an MPPT technique for maximum power extraction using INC method, in this regard, this is very old conventional technique.
Authors have implemented MPPT using Hardware in the loop, other than this, there is no significant contribution and novelty in the article.
Various MPPT techniques have been developed using various optimization algorithms with respect to those article how the performance of the article can be enhanced.
how the benefits of the techniques proposed in the articles can be overcome by proposed technique.
10.1109/JPHOTOV.2022.3150681
https://doi.org/10.3390/en15238860
https://doi.org/10.1063/1.4939531
https://doi.org/10.1007/978-81-322-2656-7_8
10.1109/ACCESS.2019.2937600
In addition there exist various mppt techniques, compare to that, what is the new innovation in the article yet to mention in detail.
Overall the quality of the work need to enhance.
Presented results are not sufficient to prove the concept. Due to this article cant be process in its present form.
Author Response
Dear reviewer thank you for all your valuable contributions and suggestions.
Please see the attachment with the responses to your comments.
Best Regards,
Sérgio André

Reviewer 3 Report
The paper is well organized and has detailed information for a novel MPPT technique based on the integral feedback of conductance, and compares it to the other widely used techniques in the literature. The authors each method's efficiency, performance, and computational needs using a HIL system. Reviewer thanks to authors for their comprehensive and well-written scientific paper.
The reviewer thinks that the authors should answer some questions for increasing the scientific value of the study.
- The authors made technical evaluations while comparing the methods in the work. Reviewer think it would be helpful to make an approximate economic comparison of methods, especially in terms of their contribution to widespread practice.
- Photovoltaic panels and energy system components are electronic equipment that must operate with high efficiency under very different physical conditions in terms of usage areas. Since the tests of the proposed model are carried out under laboratory conditions, information should be given about the working stability in such outdoor working conditions.
Author Response

(The authors gave the same response as above.)

Round 2
Reviewer 2 Report
Authors have incorporated all comments suggested by me. I dont have any other comments. Therefore, article can accept for its publication.